

# Temporal and spatial decoupling of CO$_2$ and N$_2$O soil emissions in a Mediterranean riparian forest

Sílvia Poblador [1*], Anna Lupon [1,2*], Santiago Sabaté [1,3], Francesc Sabater [1,3]

[1] Departament de Biologia Evolutiva, Ecologia i Ciències Ambientals (BEECA), Universitat de Barcelona. Av. Diagonal 643, 08028, Barcelona, Spain.

[2] Department of Forest Ecology and Management, Swedish University of Agricultural Sciences (SLU). Skogsmarksgränd 17S, 90183, Umeå, Sweden.

[3] CREAF. Campus de Bellaterra Edifici C, 08193, Cerdanyola del Vallès, Spain.

* These authors contributed equally to the development of this work.

*Correspondace to*: Sílvia Poblador (spoblador@ub.edu)

**Abstract.**

Riparian zones play a fundamental role in regulating the amount of carbon (C) and nitrogen (N) that is exported from catchments. However, C and N removal via soil gaseous pathways can influence local budgets of greenhouse gases (GHG) emissions and contribute to climate change. Over a year, we quantified soil effluxes of carbon dioxide (CO$_2$) and nitrous oxide (N$_2$O) from a Mediterranean riparian forest in order to understand the role of these ecosystems on catchment GHG emissions. In addition, we evaluated the main soil microbial processes that produce GHG (mineralization, nitrification, and denitrification) and how changes in soil properties can modify the GHG production over time and space. Mediterranean riparian soils emitted large amounts of CO$_2$ to the atmosphere (1.2 – 10 g C m$^{-2}$ d$^{-1}$), but were powerless sources of N$_2$O (0.001 – 0.2 mg N m$^{-2}$ d$^{-1}$) due to low denitrification rates. Both CO$_2$ and N$_2$O emissions showed a marked (but antagonistic) spatial gradient as a result of variations in soil moisture across the riparian zone. Deep groundwater tables fueled large soil CO$_2$ effluxes near the hillslope, while N$_2$O emissions were higher in the wet zones adjacent to the stream channel. However, both CO$_2$ and N$_2$O emissions peaked after spring rewetting events, when optimal conditions of soil moisture, temperature, and N availability favor microbial respiration, nitrification, and denitrification. Overall, our results highlight the role of riparian soils as hotspots of GHG emissions, and suggest that future alterations in hydrologic regimes can affect the microbial processes that produce GHG as well as the contribution of these systems to climate change.

**Keywords.**

greenhouse gas emissions, riparian soils, denitrification, microbial respiration, soil moisture.



## 1 Introduction

Riparian zones are hotspots of nitrogen (N) transformations across the landscape, providing a natural filter for nitrate ($NO_3^-$) transported from surrounding lands via runoff and subsurface flow paths (Hill, 1996; Vidon et al., 2010). Although interest in riparian zones has primarily been motivated by the benefits of these ecotones as effective N sinks, enhanced microbial activity in riparian landscapes can play a key role in atmospheric pollution. In temperate riparian zones, the primary N removal mechanism is denitrification, an anaerobic process whereby $NO_3^-$ is transformed to N gas or, less frequently, to nitrous oxide ($N_2O$) (Clément et al., 2002). In those cases, soil nitrification and denitrification can increase atmospheric $N_2O$ concentration by emitting up to 30 kg N ha$^{-1}$ yr$^{-1}$ (i.e. 70% of total emissions) of this powerful "greenhouse" gas (GHG) to the atmosphere (Audet et al., 2014; Cole et al., 1996; Groffman et al., 2000; Hefting et al., 2003). Moreover, the saturation of soils can support large methane ($CH_4$) fluxes that account for the 15 – 40 % of global emissions (Audet et al., 2014; Segers, 1998). Conversely, in arid or semi-arid regions, aerobic transformations involved in the oxidation of the organic matter and reduced C and N forms (i.e. respiration, mineralization, nitrification, methane oxidation) dominate the riparian biogeochemistry (Harms and Grimm, 2008). While these processes can minimize riparian $CH_4$ emissions, they can also contribute to increase atmospheric concentrations of both $N_2O$ and carbon dioxide ($CO_2$) (Batson et al., 2015). Yet, information remains scarce regarding the impact of arid and semi-arid riparian soils on the total $CO_2$ and $N_2O$ emissions from catchments.

Gas emissions from riparian soils appear to be very variable over space, reaching contradictory results concerning the potential role of riparian zones as sinks or sources of GHG emissions (Bruland et al., 2006; Groffman et al., 1992; Harms et al., 2009; Walker et al., 2002). Multiple environmental variables, such as soil temperature, moisture, and both C and N availability have been identified as key factors influencing the rate and variability of GHG exchange dynamics (Chang et al., 2014; Hefting et al., 2003; Mander et al., 2008; McGlynn and Seibert, 2003). However, all these factors tend to show strong gradients across riparian zones, which ultimately may affect the spatial pattern of GHG emissions. For instance, the riparian-hillslope edge has higher C and $NO_3^-$ concentrations compared to the near-stream area, and thus, this zone is commonly considered as a hotspot of microbial activity and GHG effluxes within riparian systems (Clément et al., 2002; DeSimone et al., 2010; Dhondt et al., 2004; Hedin et al., 1998). However, in arid or Mediterranean riparian zones, the riparian-hillslope edge is commonly water limited, which may deplete (or even inhibit) the soil microbial activity (Linn and Doran, 1984; Lupon et al., 2015). Therefore, spatial patterns of gas emissions in Mediterranean riparian zones may differ greatly from those reported in other systems, yet this sort of information still remains poorly unknown.

In addition, arid and Mediterranean regions are subjected to seasonal alterations of precipitation and temperature regimes that might affect microbial activity in riparian soils, which ultimately may difficult to upscale the relative significance of their riparian zones as GHG sources (Bernal et al., 2007; Bruland et al., 2006; Harms et al., 2009; Harms and Grimm, 2008). Increments in GHG emissions in riparian zones can occur following storms because sharp increments in soil moisture enhance denitrification, respiration, and methanogenesis rates (Casals et al., 2011; Werner et al., 2014, Harms and Grimm 2012,




Jachinte et al 2005). Conversely, GHG emissions tend to be lower in winter as a result of low temperatures and soil moisture content (Chang et al., 2014). Finally, soil $N_2O$ and $CH_4$ emissions may be low or insignificant during dry summers, while $CO_2$

emission can be enhanced due to the warmer temperatures, which induce increases in respiration rates (Batson et al., 2015; Chang et al., 2014; Lupon et al., 2016). Therefore, improved understanding of interactions among soil properties, microbial processes, and gas emissions within Mediterranean riparian zones is necessary to estimate the contribution of these ecosystems to atmospheric GHG budgets at global scale.

Despite of the importance of riparian GHG emissions for the environment, only few studies have measured simultaneously

several GHG effluxes in Mediterranean regions. Here, we aimed to evaluate patterns and controls of $CO_2$ and $N_2O$ emissions in a Mediterranean riparian forest and to assess how changes in soil properties and soil N processes across a riparian gradient can vary the gas effluxes from the hillslope edge to near-stream zones. We did not measured $CH_4$ emissions because they are considered negligible in Mediterranean systems (Batson et al., 2015; Gómez-Gener et al., 2015). We hypothesized that the studied riparian forest would emit large amounts of $CO_2$, but not of $N_2O$, to the atmosphere because aerobic soil conditions

would favor mineralization over denitrification during most of the year. Moreover, we hypothesized that soil GHG emissions would differ across the riparian gradient as a result of changes in groundwater level, soil texture, and substrate availability. We expected that C-rich soils located near the hillslope would exhibit high $CO_2$ emissions, while high $N_2O$ emissions would occur in the wet soils adjacent to the stream. Finally, we expected that temporal variability in soil moisture, temperature, and both C and N availability would drive seasonal changes in gas emissions, with peaks of $N_2O$ and $CO_2$ emissions occurring

during the wet (spring and fall) and dry (summer) seasons, respectively.

## 2 Study site

The research was conducted in a riparian forest of Font del Regàs, a forested headwater catchment (14.2 $km^2$, 500 - 1500 m above the sea level (a.s.l.)) located in the Montseny Natural Park, NE Spain (41º50'N, 2º30'E). The climate is sub-humid Mediterranean; with mean temperature ranging from 5ºC in February to 25ºC in August. In 2013, annual precipitation (1020

85 mm) was higher than long-term average (925 ± 151 mm), with most of rain falling in spring (500 mm) (Fig. 1a). Total inorganic N deposition oscillates between 15-30 kg N ha$^{-1}$ yr$^{-1}$ (period 1983-2007; Àvila and Rodà, 2012).

We selected a riparian site (~600 $m^2$, ~30 m wide) that flanked a 3$^{rd}$ order stream close to the catchment outlet (536 m a.s.l., 5.3 km from headwaters). The riparian site was divided into three zones characterized by different species composition. The near-stream zone was located adjacent to the stream (0-4 m from the stream edge) and was composed of *Alnus glutinosa* (45%

of basal area) and *Populus nigra* (33% of basal area). The intermediate zone (4-7 m from the stream edge) was composed by *P. nigra* and *Robinia pseudoacacia* (29% and 71% of basal area respectively). Finally, the hillslope zone (7-30 m from the stream edge) bordered upland forests and was composed by *R. pseudoacacia* (93% of basal area) and *Fraxinus excelsior* (7% of basal area). The three riparian zones had sandy-loam soils (bulk density = 0.9-1.1 g cm$^{-3}$), with a 5-cm deep organic layer



followed by a 30-cm deep A-horizon. The top soil layer (0-10 cm depth) was mainly composed by sands (~90%) and silts
(~7%) at the near-stream zone, whereas gravels (~16%) and sands (~80%) were the dominant particle sizes at the intermediate
and hillslope zones. During the study period, groundwater level (GWL) averaged -54 ± 14 cm below the soil surface (b.s.s.)
at the near-stream zone, and decreased to -125 ± 4 and -358 ± 26 cm b.s.s. at the intermediate and hillslope zones, respectively
(Fig. 1b).

# 3 Materials and methods

## 3.1 Field sampling

We delimited five plots (1 x 1 m) within each riparian zone (near-stream, intermediate and hillslope). During the year 2013,
soil physicochemical properties, soil N processes, and gas emissions were measured in each plot every 2-3 months in order to
cover a wide range of moisture and temperature conditions. On each sampling month, one soil sample (0-10 cm depth,
including O- and A- horizons) was collected randomly from each plot to analyze soil physicochemical properties. Soil samples
were taken with a 5-cm diameter core sampler and placed gently into plastic bags after carefully removing the litter layer. Close
to each soil sample, we performed *in situ* soil incubations to measure soil net N mineralization (NNM) and net nitrification (NN)
rates (Eno, 1960). For this purpose, a second soil core (0-10 cm depth) was taken, placed in a polyethylene bag, and buried at
the same depth. Soil incubations were buried 4 days and then removed from the soil.

Gas emissions and denitrification rates were measured simultaneously and during four consecutive days (i.e. during the entire
soil incubation period) in order to facilitate the direct comparison between microbial rates and gas fluxes. Soil $CO_2$ effluxes
were measured by using a SRC-1 soil chamber attached to an EGM-4 portable infrared gas analyzer (IRGA) (PP Systems,
Amesbury, MA). The EGM-4 has a measurement range of 0-2000 ppm ($\mu$mol mol$^{-1}$), with an accuracy of 1% and a linearity
of 1% throughout the range. $CO_2$ emissions rates were calculated as the amount of $CO_2$ accumulated in the head-space of the
EGM-4 chamber after an incubation time of c.a. 120 s. *In situ* denitrification rates (DNT) and $N_2O$ emissions were measured
using closed cylinder (0.37 L) and open cylinder (0.314 m$^2$) chambers, respectively. For DNT analyses, an intact soil core (0-
10 cm depth) was introduced in the chamber, closed with a rubber serum stopper, amended with acetone-free acetylene to
inhibit the transformation of $N_2O$ to $N_2$ (10% v/v atmosphere), and placed at the same depth. For $N_2O$ analysis, chambers were
placed directly on the soil and no special treatment was carried out. Gas samples for both DNT and $N_2O$ chambers were taken
at the same time (0h, 1h, 2h, and 4h of incubation) with a 20-mL syringe and stored in evacuated tubes. All soil and gas samples
were kept at < 4ºC until laboratory analysis (< 24 h after collection).

Soil physical properties were measured within each plot simultaneously to gas emissions. Volumetric soil moisture (SWC, %)
(5 replicates per plot) and soil temperature (Tsoil, ºC) (1 replicate per plot) were measured at 10-cm depth by using a time-
domain reflectometer sensor (HH2 Delta-T Devices Moisture Meter) and a temperature sensor (CRISON 25), respectively.
Soil pH and reduction potential (Eh, mV) (1 replicate per plot) were measured at 0-10 cm depth by water extraction (1:2.5 v/v)



using a Thermo-Scientific ORION sensor (STAR 9107BNMD). Although Eh measures performed by water extraction may not be as accurate as other field technics, these values have been previously used as a good proxy of the soil redox potential (Yu and Rinklebe, 2013).

### 3.2 Laboratory analyses

Pre-incubation soil samples were oven dried at 60ºC, sieved, and the fraction < 2 mm was used for measuring soil chemical properties. The relative soil organic matter content (SOM, %) was measured by loss on ignition (450ºC, 4 h). Total soil C and N contents were determined on a gas chromatograph coupled to a TCD detector after combustion at 1000ºC at the Scientific Technical Service of the University of Barcelona.

To estimate microbial N processes, we extracted 5 g of pre- and post- incubation field-moist soil samples with 50 ml of 2 M KCl (1g : 10ml, ww : v; 1 h shacking at 110 r.p.m. and 20ºC). The supernatant was filtered (Whatman GF/F 0.7 μm pore
diameter) and analyzed for ammonium ($NH_4^+$) and nitrate ($NO_3^-$). $NH_4^+$ was analyzed by the salicilate-nitropruside method (Baethgen and Alley, 1989) using a spectrophotometer (PharmaSpec UV-1700, SHIMADZU). $NO_3^-$ was analyzed by the cadmium reduction method (Keeney and Nelson, 1982) using a Technicon Autoanalyzer (Technicon, 1987). For each pair of samples, NNM and NN were calculated as the differences between pre- and post-incubations values of inorganic N ($NH_4^+$ and $NO_3^-$) and $NO_3^-$, respectively (Eno, 1960). Pre-incubation $NH_4^+$ and $NO_3^-$ concentrations were further used to calculate the
availability of dissolved inorganic nitrogen (DIN) in riparian soils.

To estimate DNT and natural $N_2O$ emissions, we analyzed the $N_2O$ of all gas samples using a gas chromatograph (Agilent Technologies, 7820A GC System). Both DNT and $N_2O$ emissions rates were calculated as the amount of $N_2O$ accumulated in the head-space of the chamber after 4h of incubation. In addition, we measured the denitrification enzyme activity (DEA) for 3 soil cores of each riparian zone to determine the factors limiting denitrification. For each soil core, four sub-samples (20 g
of fresh soil) were placed into 125-ml glass jars containing different treatments. The first jar ($DEA_{MQ}$) contained Milli-Q water (20 ml) to test anaerobiosis limitation. The second jar ($DEA_C$) was amended with glucose solution (4 g glucose kg soil$^{-1}$) to test C limitation. The third jar ($DEA_{NO3}$) was amended with nitrate solution (72.22mg $KNO_3$ kg soil$^{-1}$) to test N limitation. Finally, the fourth jar ($DEA_{C+NO3}$) was amended with both nitrate and glucose solutions (4 g glucose kg soil$^{-1}$ and 72.22mg $KNO_3$ kg soil$^{-1}$) to test simultaneously C and N limitation. All jars were capped with rubber serum stoppers, made anaerobic
by flushing $N_2$, and amended with acetone-free acetylene (10% v/v) (Smith and Tiedje, 1979). Gas samples were collected after 4 h and 8 h of incubation and analyzed following the same procedure of field DNT samples. DEA rates were calculated as the rate of $N_2O$ accumulation in the headspace.

### 3.3 Data analysis

Statistical analyses were carried out using the package *lmer* and *pls* of R 2.15.1 statistical software (R Core Team, 2012). We
performed mixed-model analysis of variance (ANOVA) to test differences in soil properties, microbial N processes, and gas



emissions across riparian zones and seasons. We used riparian zone and season as fixed effects, and plot (nested within riparian zones) as a random effect. When multiple samples were taken within a plot (soil physical properties, DNT, and gas emissions), the ANOVA was performed on plot means, with n = 75 (5 plots x 3 zones x 5 dates). For each model, post-hoc Tukey contrasts were used to test which zones or seasons differed from each other. In all cases, residuals were tested for normality using a Shapiro-Wilk test, and homogeneity of variance was examined visually by plotting the predicted and residual values. In those cases that the normality assumption was unmet, data was log transformed. In all analyses, differences were considered significant when $p < 0.05$.

We used partial least squares regression (PLS) to explore how soil properties, C and N availability, GWL, and soil N processes predict variation in $CO_2$ and $N_2O$ emissions. PLS identifies the relationship between independent (X) and dependent (Y) data matrices through a linear, multivariate model; and produces latent variables (PLS components) representing the combination of X variables that best describe the distribution of observations in 'Y space' (Eriksson et al., 2006). We determined the goodness of fit ($R^2Y$) and the predictive ability ($Q^2Y$) of the model by comparing modeled and actual Y observations through a cross-validation process. Each model was refined by iteratively removing variables that had non-significant coefficients in order to minimize the model overfitting (i.e. low $Q^2Y$ values) as well as the multicollinearity of the explanatory variables (i.e. variance inflation factor (VIF) < 5). Furthermore, we identified the importance of each X variable by using variable importance on the projection (VIP) scores, calculated as the sum of square of the PLS weights across all components. VIP values > 1 indicate variables that are most important to the overall model (Eriksson et al., 2006). In all PLS models, data was ranked and centered prior analysis.

## 4 Results

### 4.1 Spatial pattern of soil properties, microbial rates, and gas emissions

During the study period, all riparian zones had similar mean soil temperature (11 – 12ºC), pH (6 – 7) and redox potential (170 – 185 mV) (Table 1). However, soil moisture exhibited strong differences across riparian zones (Table 2), with the near-stream zone holding wetter soils than the intermediate and the hillslope zones (Table 1). There were significant differences in most of soil chemical properties (Table 1, Table 2). Both SOM and soil C and N content were 2-fold lower in the near-stream zone than in the intermediate and hillslope zones, though all zones exhibited similar C:N ratios (CN = 14). Moreover, DIN concentrations ($NH_4^+$ and $NO_3^-$) were from 2- to 5-fold lower for the near-stream zone than for the other two zones.

On annual basis, NNM averaged $0.14 \pm 0.40$, $0.39 \pm 1.23$, and $0.22 \pm 1.03$ mg N $kg^{-1}$ $d^{-1}$ at the near-stream, intermediate, and hillslope zones, respectively. Mean annual NN rates were close to NNM, averaging $0.17 \pm 0.38$, $0.25 \pm 0.69$, and $0.28 \pm 0.73$ mg N $kg^{-1}$ $d^{-1}$ at the near-stream, intermediate, and hillslope zones, respectively. There were no significant differences in mean annual NNM and NN rates among riparian zones (in both cases: mixed-model ANOVA test, $F > F_{0.05}$, $p > 0.05$). Mean annual DNT was higher at the near-stream zone ($2.69 \pm 5.30$ mg N $kg^{-1}$ $d^{-1}$) than at the intermediate ($0.72 \pm 1.85$ mg N $kg^{-1}$ $d^{-1}$) and



hillslope ($0.76 \pm 1.59$ mg N kg$^{-1}$ d$^{-1}$) zones (mixed-model ANOVA test, F = 4.33, p = 0.038). However, potential DNT rates were lower in the near-stream zone ($0.3 - 0.6$ mg N kg$^{-1}$ d$^{-1}$) compared to intermediate ($1.0 - 2.4$ mg N kg$^{-1}$ d$^{-1}$) and hillslope ($1.3 - 3.8$ mg N kg$^{-1}$ d$^{-1}$) zones (Table 3).

Natural $CO_2$ and $N_2O$ emissions differed among riparian zones, yet they showed opposite spatial patterns. Near-stream zone exhibited lower $CO_2$ emissions ($318 \pm 195$ mg C m$^{-2}$ h$^{-1}$) compared to the intermediate ($472 \pm 298$ mg C m$^{-2}$ h$^{-1}$) and hillslope ($458 \pm 308$ mg C m$^{-2}$ h$^{-1}$) zones (mixed-model ANOVA test, F = 7.08, p = 0.009). Conversely, near-stream zone showed higher $N_2O$ emissions ($0.035 \pm 0.022$ mg N m$^{-2}$ h$^{-1}$) than the other two zones (intermediate = $0.032 \pm 0.025$ mg N m$^{-2}$ h$^{-1}$; hillslope = $0.022 \pm 0.012$ mg N m$^{-2}$ h$^{-1}$ ) (mixed-model ANOVA test, F = 7.31, p = 0.008).

**4.2 Temporal pattern of soil properties, microbial rates, and gas emissions**

During the study period, there was a marked seasonality in most of soil physical properties, except for pH and Eh, which did not show any temporal pattern (Table 2). Soil moisture showed a marked seasonality, though it differed among riparian zones (Table 2, "zone x season"). In the intermediate and hillslope zones, soil moisture was maxima in November and minima in August, while the near-stream soils were wetter during both spring (April-June) and autumn (November) (Fig. 2a). Conversely,
soil temperature showed similar seasonality but opposite values in all riparian zones (Table 2), with a maxima in summer (August) and minima in winter (February) (Fig. 2b). Soil chemical properties (SOM and both soil C and N content) did not show any seasonal trend, but all riparian zones exhibited lower C:N ratios in February compared to the other seasons (Fig. 2c). There was no seasonality in soil $NH_4^+$ concentrations at any riparian zone (Table 2). However, soil $NO_3^-$ concentrations showed a marked temporal pattern, yet it differed among riparian zones (Table 2, "zone x season"). The highest soil $NO_3^-$
concentrations occurred in February at both the near-stream and hillslope zones, but in June-August at the intermediate zone (Fig. 2d).

Soil N processes showed similar seasonal patterns in all riparian zones (in all cases: $F_{date} < F_{0.05}$, $F_{interaction} > F_{0.05}$). Both NNM and NN rates were higher in April than February, June, and November (Fig. 3a and 3b), while DNT rates were higher in April and June compared to the rest of the year (Fig. 3c). In April, both NNM and NN rates differed across riparian zone, with
higher rates in the intermediate zone than in the near-stream one. NNM rates also differed in August, when the intermediate zone exhibited 2-fold higher rates than the other two zones. Finally, DNT was higher at the near-stream than at the other two zones in both June and August.

Natural gas emissions showed a clear seasonal pattern (in both cases: mixed-model ANOVA test, $F_{date} < F_{0.05}$, p < 0.001), yet it differed between $CO_2$ and $N_2O$ emissions. In all zones, $CO_2$ emissions were maxima in June and minima in February (Fig.
4a), while highest $N_2O$ emission rates occurred in April and lowest in both February and August (Fig. 4b). In spring (April and June), $CO_2$ emissions were higher at the intermediate and hillslope zones compared to the near-stream one (Fig. 4a). Moreover, the near-stream zone showed higher $N_2O$ emissions than the hillslope zone in February, April, and June (Fig. 4b).



### 4.3 Relationship between soil properties, microbial processes, and gas emissions

PLS models extracted two components that explained the 71% and the 40% of the variance in $CO_2$ and $N_2O$ emissions,

respectively (Table 4). The model predictability was high for $CO_2$ ($Q^2Y = 0.66$), but weak for $N_2O$ ($Q^2Y = 0.34$). Moreover, PLS models identified few variables as key predictors of GHG emissions (VIF < 2, VIP > 0.8), yet these variables differed between $CO_2$ and $N_2O$ emissions (Table 4). Soil temperature (PLS coefficient [coef] = +0.60), and soil moisture (coef = -0.24) explained most of the variation in $CO_2$ emissions (Fig. 5a). Conversely, variations in $N_2O$ emissions were primarily related to changes in denitrification rates (coef = +0.45), soil moisture (coef = +0.21) and, to less extent, groundwater level (coef = -

0.16) (Fig. 5b).

## 5 Discussion

### 5.1 Daily soil GHG emissions

Mean daily emissions of $CO_2$ found in the present study (1.2 – 10 g C m$^{-2}$ d$^{-1}$) were generally high, especially during spring and summer months. These soil $CO_2$ emissions were higher than those reported for wetlands or riparian zones in temperate

and boreal systems (0.2 – 4.8 g C m$^{-2}$ d$^{-1}$) (Batson et al., 2015; Bond-Lamberty and Thomson, 2010; Mander et al., 2008), although similar values have been reported in some dry forested wetlands of Europe and North America (Harms and Grimm, 2008; Oertel et al., 2016). These substantially high $CO_2$ emissions observed in Font del Regàs may be attributed to high in-situ microbial respiration associated with relatively moist and SOM enriched soils (Mitsch and Gosselink, 2007; Pacific et al., 2008; Stern, 2006). In agreement, previous studies have reported that microbial heterotrophic respiration can be an important

contributor (> 60%) to $CO_2$ effluxes in semi-arid and Mediterranean riparian zones (Harms and Grimm, 2012; McLain and Martens, 2006). However, the absence of a relationship between soil N processes and $CO_2$ emissions suggests that other microbial heterotrophic processes rather than N mineralization may drive $CO_2$ emissions in this Mediterranean riparian zone, and thus, soil N mineralization may be not a good descriptor of bulk organic matter mineralization. Moreover, plant roots respiration and methane oxidation can increase the $CO_2$ emissions in riparian soils with deep groundwater tables such as in

Font del Regàs (Chang et al., 2014). Accordingly, extremely low or negative $CH_4$ emissions (-0.06 – 0.42 mg C m$^{-2}$ d$^{-1}$) have been reported in dry riparian zones, which only exhibited high values when soils saturated during flood events (Batson et al., 2015; Harms and Grimm, 2012; Jacinthe et al., 2015). During the study period, riparian soils were never saturated, and thus, we expect a negligible contribution of our riparian soils to global $CH_4$ emissions.

Conversely, $N_2O$ emissions of our riparian site (0.001 – 0.2 mg N m$^{-2}$ d$^{-1}$) were relatively low during the whole year. Similar

$N_2O$ emissions were reported in other water limited riparian forests that are rarely flooded (-0.9 – 0.39 mg m$^{-2}$ d$^{-1}$; Bernal et al., 2003; Harms and Grimm, 2012; Vidon et al., 2016), yet these values were, on average, much lower than those found in temperate riparian zones (0 – 54 mg N m$^{-2}$ d$^{-1}$; Burgin and Groffman, 2012; Hefting et al., 2003; Mander et al., 2008). In Font del Regàs, low gas emissions may be partially attributed to low denitrification rates, as we found an intimate link between this microbial process and $N_2O$ emissions. Likely, the inhibition of denitrification was caused by soil dryness because, at our site,





riparian groundwater table usually flowed well below the soil surface (> 50 cm b.s.s.), and thus, optimal moisture conditions
for denitrification (SWC > 60%; Pinay et al., 2007) were infrequent even in spring, when large rainfall events occurred. Yet,
area-specific denitrification rates ($0.1 - 0.3$ mg N m$^{-2}$ d$^{-1}$) were one order of magnitude higher than soil $N_2O$ emissions,
suggesting that $N_2$ rather than $N_2O$ was the major product yielded by denitrification in our riparian site. Additionally, other
processes such as nitrification or nitrate ammonification can contribute to $N_2O$ emissions (Baggs, 2008; Hefting et al., 2003).
However, it seems unlikely that nitrification could account for the observed $N_2O$ emissions because no relationship was found
between net nitrification rates and $N_2O$ emissions, while relatively oxic conditions (Eh > 100) and low C:N ratios (C:N < 20)
suggest low nitrate ammonification in riparian soils (Schmidt et al., 2011). Currently, the influence of soil denitrification on
$N_2O$ emissions in riparian zones is still under debate (Giles et al., 2012). Nonetheless, our results suggest that performing
simultaneous measurements of different soil N processes can contribute to disentangle the mechanisms underlying net $N_2O$
emissions in riparian areas.

There is still little research available on whether processes occurring in riparian soils can have any implication at larger spatial
scales and how the mechanisms underlying such GHG emissions can ultimately modify catchment GHG fluxes. Our results
suggest that Mediterranean riparian soils can be a powerless source of $N_2O$ to the atmosphere because daily $N_2O$ emissions
equaled, on average, to 8.9 mg C m$^{-2}$ d$^{-1}$ (based on $N_2O$:$CO_2$ radiative warming equivalent of 1:298; Forester et al., 2007).
However, the $CO_2$ effluxes recorded in our Mediterranean riparian soils were much higher than those found in their surrounding
uplands ($0.1 - 3.3$ g C m$^{-2}$ d$^{-1}$; Barba et al., 2016; Kesik et al., 2005) and streams ($0.2 - 5.5$ g C m$^{-2}$ d$^{-1}$; Gómez-Gener et al.,
2015; von Schiller et al., 2014). When accounting for all GHG ($CO_2$ + $N_2O$), our study suggest that riparian soils can emit
between $438 - 3650$ g C m$^{-2}$ yr$^{-1}$. Assuming that GHG emissions ($CO_2$ + $N_2O$) from upland evergreen oak and beech soils
(54% and 38% of the catchment, respectively) are similar to other Mediterranean regions (oak: $19 - 1240$ g C m$^{-2}$ yr$^{-1}$; Asensio
et al., 2007; Inclán et al., 2014; beech: $214 - 1182$ g C m$^{-2}$ yr$^{-1}$; Guidolotti et al., 2013; Kesik et al., 2005), then riparian soils
can contribute 16- 22% of total catchment soil GHG emissions despite occupying a small area of the catchment (6%). Although
these estimates are rough, our results clearly pinpoint that riparian soils can be potential hot spots of GHG emissions within
Mediterranean catchments. These findings contrast with the common knowledge that water limited riparian soils are powerless
GHG sources to the atmosphere (Bernal et al., 2007; Vidon et al., 2016) and stress the importance of simultaneously consider
several GHG emissions (i.e. $CO_2$, $N_2O$, $CH_4$) to get a whole picture of the role of riparian soils in climate change.

**5.2 Spatio-temporal variations of GHG emissions**

Fluxes of GHG from riparian soils display a high degree of spatial variability due to heterogeneity in soil properties (Groffman
et al., 1998; van den Heuvel et al., 2009; Hill et al., 2000). In our riparian plot, soil moisture gradually decreased from the
near-stream zone to the hillslope edge as a result of changes in groundwater level and soil texture. Moreover, we found larger
amounts of C and N available in those soils located far from the stream channel than in the near-stream zone, maybe due to
the effect of flood events that occur in these zones changing near-stream soil chemical properties (Jolley et al., 2010). Based





on soil properties, we expected that the hillslope zone would exhibit the greatest $CO_2$ emissions due to higher SOM availability. Accordingly, we found higher $CO_2$ effluxes at the intermediate and hillslope zones than at the near-stream zone. Yet, our results suggest that such gradient did not rely on substrate supply because neither SOM, C, nor N availability were selected in the PLS model. Conversely, $CO_2$ emissions in our riparian plot were negatively correlated with soil wetness, suggesting that as soils become less moist and more aerated, oxidizing aerobic respiration increases, ultimately stimulating $CO_2$ production (Muller et al., 2015). Moreover, the deep groundwater table in the hillslope zone can increase the volume of aerated soil, which can increase the area-specific soil $CO_2$ emissions near the hillslope edge (Chang et al., 2014). In agreement, increasing $CO_2$ emissions from wet to dry zones has been reported in other wetlands and riparian forests (Batson et al., 2015; Morse et al., 2012; Welti et al., 2012), pinpointing that variations in riparian hydrology can play a fundamental role in GHG emissions from riparian soils.

As expected, $N_2O$ fluxes showed a clear pattern across the riparian plot, with $N_2O$ emissions being higher in the near-stream zone than in the other two zones. Such spatial pattern was different from those found in other riparian forests, where higher $N_2O$ emissions occurred in the hillslope edge zone as a result of higher C and $NO_3^-$ availability (DeSimone et al., 2010; Dhondt et al., 2004; Hedin et al., 1998). As occurred for $CO_2$ emissions, we suggest that the observed spatial pattern may be as a result of changes in water availability across the riparian zone. In the near-stream zone, relatively moist conditions (SWC = 30 – 40%) can promote denitrification rates (Pinay et al., 2007), but also induce greater $N_2O$ production by preventing the reduction $N_2O$ to $N_2$ (Giles et al., 2012). Conversely, dry soils (SWC = 10 – 25%) can limit denitrification in the intermediate and hillslope zones (Linn and Doran, 1984; Pinay et al., 2007), thus decreasing the overall $N_2O$ emissions in these areas. This former idea is further supported by DEA results, which showed that, after adding water, denitrification rates were similar to those observed in the field for the near-stream zone, but increase by 3-4 fold in the other two zones. Furthermore, our DEA results pinpoint that the riparian-hillslope edge can be a potential hot spot of N removal within Mediterranean riparian zones, because high denitrification rates were observed when favorable conditions (i.e. high water, C, and N availability) occurred.

Soil $CO_2$ and $N_2O$ emissions also varied temporally. Similarly to other dry riparian zones, $CO_2$ emissions were the highest in late-spring and the lowest in winter (Harms and Grimm, 2012; Morse et al., 2012). As previously reported, such intra-annual variations were strongly dependent on seasonal changes in soil temperature because it was the most influential environmental variable in the PLS model (Chang et al., 2014; Morse et al., 2012; Wickland et al., 2010). Therefore, cold temperatures (< 4°C) probably limited soil respiration in riparian forests during winter; while warm conditions (> 15°C) stimulated soil $CO_2$ emissions in June and August (Emmett et al., 2004; Suseela et al., 2012; Teiter and Mander, 2005). However, lower $CO_2$ emissions than expected for temperature dynamics were reported in summer at the intermediate and hillslope zones, likely because extreme dryness (SWC < 20%) limited respiration rates during such period (Chang et al., 2014; Goulden et al., 2004; Wickland et al., 2010). Although the mechanisms by which soil dryness may affect C demand are still poorly understood, suppressed microbial respiration in summer can be attributed to a disconnection between microbes and resources (Belnap et





al., 2005; Davidson et al., 2006), decreases in photosynthetic and exo-enzimatic activities (Stark and Firestone, 1995; Williams
et al., 2000), or a relocation of the invested energy on growth (Allison et al., 2010). Altogether, these results suggest that soil
moisture may be as important as soil temperature in order to understand soil $CO_2$ effluxes, and therefore, future warmer
conditions may not fuel higher $CO_2$ emissions, at least in those regions experiencing severe water limitation.

In addition, a strong seasonality in $N_2O$ emission was observed in all riparian zones. High rates of $N_2O$ effluxes occurred in
spring, which could be likely driven by increments in soil moisture after rainfall (or flood) events (DeSimone et al., 2010;
Jacinthe et al., 2009; Scholes et al., 1997). Pulses of $N_2O$ emissions short-after rewetting events can reflect the microbial use
of $NO_3^-$ that has been accumulated during dry antecedent periods. In our riparian site, soil $NO_3^-$ concentrations were high during
winter, when cold temperatures and low SWC probably limited both denitrification rates and gas effluxes (Chang et al., 2014;
Hefting et al., 2004; Pinay et al., 2007). Moreover, our results further suggest that soil microbial activity was stimulated during
spring rewetting events because both nitrification and denitrification rates were maxima in April, when large precipitation
events (400 mm) raised the groundwater level and increased SWC at the whole riparian plot. However, the studied riparian
soils remained unsaturated during most of spring, and thus, both the production and diffusion of $N_2O$ remained high (Scholes
et al., 1997). This idea agrees with our PLS model, which suggests that denitrification, soil moisture, and $NO_3^-$ concentrations
are the key variables explaining $N_2O$ variability in our riparian soil.

Nevertheless, our results also pinpoint that the temporal and spatial patterns of $N_2O$ emissions are difficult to predict. For
instance, we expected large $N_2O$ emissions following rains in November because, similarly to spring, environmental conditions
(i.e. high SWC, mid soil temperatures, and increments in soil $NO_3^-$ concentrations during the antecedent dry summer) should
enhance microbial activity. However, both nitrification and denitrification rates were low in November, which ultimately
decrease $N_2O$ compared to April. Possibly, low denitrification rates in fall may be attributed to an increase in N demand
following large C inputs from litterfall (Guckland et al., 2010; Lupon et al., 2016). Moreover, leaf litter from *R. pseudoacacia*,
the main tree species in our study site, holds a high lignin content (Castro-Díez et al., 2009; Yavitt et al., 1997), which might
enrich the riparian soil with phenolic compounds and ultimately limit denitrification rates (Bardon et al., 2014). These results
suggest that the response of the microbial community to changes in water availability may depend on the interplay of additional
ecosystem factors not included in this study. Therefore, we propose that simultaneous measurements of environmental factors,
soil microbial activity, and microbial structure should be performed in order to get a complete comprehension of GHG
emissions in Mediterranean riparian zones.

## 6 Conclusions

Mediterranean riparian zones are dynamic systems that undergo spatial and temporal shifts in biogeochemical processes due
to changes in both soil water and substrate availability. From these observations, some authors have proposed that the
contribution of Mediterranean riparian zones on catchment budgets and exports may differ greatly from those observed in



temperate systems (Lupon et al., 2016), yet their contribution to atmospheric pollution is still under debate. In a first attempt to simultaneously quantify $CO_2$ and $N_2O$ emissions from Mediterranean riparian soils, we showed that they can emit large amounts of GHG to the atmosphere in form of $CO_2$, but not as $N_2O$. In addition, our results clearly illustrate a strong linkage between riparian hydrology and the microbial processes that produce GHG. Deep groundwater tables fueled large respiration rates and soil $CO_2$ effluxes in the relatively dry soils near the hillslope, while both denitrification and $N_2O$ effluxes were higher

in the wet zones located near the stream channel. As occurred at spatial scale, temporal patterns of $CO_2$ and $N_2O$ emissions were decoupled during most of the year. However, effluxes of both GHG peaked after rainfall events in spring, when optimal conditions of soil moisture, temperature, and N availability favor microbial respiration, nitrification, and denitrification. Overall, our study highlights the potential of Mediterranean riparian soils to be hotspots of GHG emissions within catchments.

**Author contributions**

Sílvia Poblador, Santiago Sabaté, and Francesc Sabater designed the experiment. Sílvia Poblador and Anna Lupon carried them out. Sílvia Poblador performed all laboratory analysis. Anna Lupon and Sílvia Poblador analyzed the data set and prepared the manuscript, with contributions from Santiago Sabaté and Francesc Sabater.

**Aknowledgments**

We are thankful to Ada Pastor and Lídia Cañas for their invaluable assistance in the field. Special thanks are extended to Núria
Catalán for helpful comments on an earlier version of the manuscript. Financial support was provided by the Spanish Government through the projects MONTES-Consolider (CSD2008-00040-MONTES), MEDFORESTREAM (CGL2011-30590), and MEDSOUL (CGL2014-59977-C3-2). Sílvia Poblador was supported by a FPI PhD fellowship from the Spanish Ministry of Economy and Competitiveness (BES-2012-054572). Anna Lupon was supported by a Kempe Foundation post-doctoral grant (Sweden) and the MEDSOUL project. We also thank site cooperators, including Vichy Catalan and the Catalan
Water Agency (ACA) for permission to sample at the Font del Regàs catchment. Sílvia Poblador, Anna Lupon, Santiago Sabaté, and Francesc Sabater are members of the research group FORESTREAM (AGAUR, Catalonia 2014SGR949).





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





## Tables

**Table 1.** Mean annual values (± standard deviation) of soil water content (SWC), soil temperature (Tsoil), soil pH, soil redox capacity (Eh), soil organic matter (SOM), soil molar C:N ratio, soil carbon (C) and nitrogen (N) content, and soil ammonium ($NH_4^+$) and nitrate ($NO_3^-$) concentrations for the three riparian zones. For each variable, 570 different letters indicate statistical significant differences between riparian zones (*post-hoc* Tukey HSD test, $p < 0.05$).

|  | *Near-stream* | *Intermediate* | *Hillslope* |
|---|---|---|---|
| **SWC (%)** | 29.58 ± 7.55 [A] | 19.36 ± 6.00 [B] | 19.81 ± 6.24 [B] |
| **Tsoil (ºC)** | 11.37 ± 5.39 [A] | 11.82 ± 5.90 [A] | 12.01 ± 6.34 [A] |
| **Eh** | 170 ± 111 [A] | 184 ± 103 [B] | 184 ± 95 [C] |
| **pH** | 6.66 ± 0.42 [A] | 6.31 ± 0.50 [A] | 6.68 ± 0.53 [A] |
| **SOM (%)** | 4.41 ± 0.71 [A] | 7.98 ± 2.88 [B] | 9.53 ± 1.99 [C] |
| **C:N ratio** | 14.25 ± 3.64 [A] | 14.09 ± 1.78 [A] | 13.63 ± 1.18 [A] |
| **C (mg kg$^{-1}$)** | 2004 ± 1038 [A] | 4007 ± 1785 [B] | 4923 ± 1428 [B] |
| **N (mg kg$^{-1}$)** | 160 ± 44 [A] | 330 ± 135 [B] | 418 ± 107 [C] |
| **$NH_4^+$ (mg N kg$^{-1}$)** | 1.88 ± 1.21 [A] | 5.58 ± 3.48 [B] | 3.90 ± 2.07 [B] |
| **$NO_3^-$ (mg N kg$^{-1}$)** | 0.75 ± 0.58 [A] | 4.66 ± 4.25 [B] | 5.30 ± 4.20 [B] |





**Table 2.** Results from the mixed-model analysis of variance (ANOVA) showing the effects of riparian zones and seasons on soil water content (SWC), soil temperature (Tsoil), soil pH, soil redox capacity (Eh), soil organic matter (SOM), soil molar C:N ratio, soil carbon (C) and nitrogen (N) content, and soil ammonium ($NH_4^+$) and nitrate ($NO_3^-$) concentrations. Plot was treated as a random effect in the model whereas riparian zones, seasons and their interactions were considered fixed effects. Values are *F*-values and the p-values are shown in brackets. P-values <
0.05 are shown in bold.

|  | *Riparian Zone* | *Seasons* | *Zone × Seasons* |
|---|---|---|---|
| **SWC** | **18.6 [< 0.001]** | **100 [< 0.001]** | **13.6 [< 0.001]** |
| **Tsoil** | 0.33 [0.721 ] | **2117 [< 0.001]** | 0.42 [0.906] |
| **pH** | 1.97 [0.182] | 2.43 [0.060] | 2.73 [0.052] |
| **Eh** | 1.34 [0.247] | 3.53 [0.062] | 1.88 [0.084] |
| **SOM** | **27.8 [< 0.001]** | 2.77 [0.053] | 1.62 [0.144] |
| **C:N ratio** | 0.99 [0.400] | **10.9 [< 0.001]** | 1.72 [1.118] |
| **C** | **27.1 [< 0.001]** | 1.86 [0.132] | 0.77 [0.630] |
| **N** | **39.7 [< 0.001]** | 1.22 [0.311] | 0.63 [0.746] |
| **$NH_4^+$** | **12.4 [0.001]** | 2.71 [0.051] | 1.52 [0.176] |
| **$NO_3^-$** | **22.4 [< 0.001]** | **5.63 [< 0.001]** | **4.09 [< 0.001]** |

Zone = near-stream, intermediate, hillslope.
Season = February, April, June, August and November.





**Table 3.** Mean values (± standard deviation) of potential denitrification rates (in mg N kg$^{-1}$ d$^{-1}$) after anoxia (DEA$_{MQ}$), carbon addition (DEA$_C$), nitrogen addition (DEA$_{NO3}$) and carbon and nitrogen addition (DEA$_{C+NO3}$) treatments for the three riparian zones during the study period. For each zone, different letters indicate statistical significant differences between treatments (*post-hoc* Tukey HSD test, n = 15, p < 0.01).

| | Potential DNT rates (mg N kg$^{-1}$ d$^{-1}$) | | | |
| --- | --- | --- | --- | --- |
| | **DEA$_{MQ}$** | **DEA$_C$** | **DEA$_{NO3}$** | **DEA $_{C+NO3}$** |
| *Near-stream* | 0.31 ± 0.41$^A$ | 0.26 ±0.27$^A$ | 0.42 ± 0.42$^A$ | 0.63 ± 0.85$^A$ |
| *Intermediate* | 1.01 ± 1.12$^A$ | 1.88 ± 1.59$^A$ | 2.28 ± 3.57$^A$ | 2.40 ± 2.45$^A$ |
| *Hillslope* | 1.34 ± 1.33$^A$ | 2.35 ± 1.97$^{AB}$ | 1.73 ± 1.43$^{AB}$ | 3.82 ± 2.78$^B$ |



**Table 4.** Summary of the partial least squares (PLS) models produced for $CO_2$ and $N_2O$ emissions at the riparian site (n = 75). Values are the coefficients from PLS models which describe the relationship (direction and relative strength) between explanatory variables and gas emissions. The variance inflation factor (VIF) of each explanatory variable, indicative of collinearity, are shown in brackets. Bold values indicate the most influencing variables (variable importance in the projection (VIP) >1.0).

|  | *X-variable* | *Acronym* | *$CO_2$* | *$N_2O$* |
|---|---|---|---|---|
| **Soil Properties** | Water content (%) | SWC | **-0.235 [1.72]** | **0.205 [1.32]** |
|  | Groundwater level (cm b.s.s.) | GWL | **---** | -0.157 [1.24] |
|  | Temperature (C) | Tsoil | **0.599 [1.45]** | **---** |
|  | pH | pH | --- | --- |
|  | Redox potential (mV) | Eh | **---** | **---** |
|  | Bulk density (g cm$^{-1}$) | BD | --- | **---** |
|  | Coarse texture (%) | % Sand | --- | **---** |
|  | Organic matter (%) | SOM | --- | --- |
|  | Total Carbon | C | --- | --- |
|  | Total Nitrogen | N | --- | --- |
|  | Molar C:N ratio | C:N ratio | --- | --- |
|  | Ammonium | $NH_4^+$ | 0.167 [1.61] | --- |
|  | Nitrate | $NO_3^-$ | 0.066 [1.80] | -0.060 [1.47] |
| **Soil N processes** | Net N Mineralization | NNM | --- | --- |
|  | Net Nitrification | NN | --- | --- |
|  | Denitrification | DNT | --- | **0.449 [1.09]** |
| **R²Y** |  |  | **0.71** | **0.40** |
| **Q²Y** |  |  | **0.66** | **0.34** |





## Figures

**Figure 1.** Temporal pattern of (a) mean monthly precipitation and (b) biweekly groundwater level at the studied riparian site during the year 2013. Circles are mean values of groundwater level at the near-stream (white), intermediate (grey), and hillslope (black) zones. Precipitation data was obtained from a meteorological station located at ca. 300 m from the studied riparian site. At each riparian zone, groundwater level was measured in 3 PVC piezometers (32-mm diameter, 1–3 m long) with a water level sensor (Eijkelkamp 11.03.30).

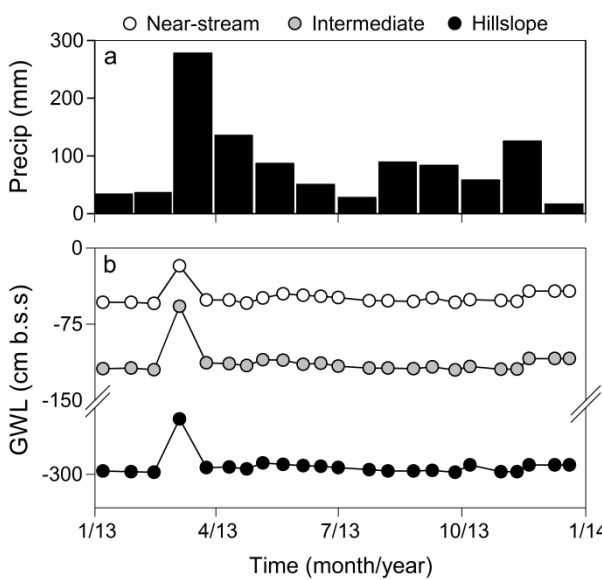






**Figure 2.** Temporal pattern of (a) soil water content (SWC), (b) soil temperature (Tsoil), (c) soil C:N molar ratio (C:N ratio), and (d) soil nitrate concentration ($NO_3^-$) at 10-cm depth. Data is shown for the near-stream (white), intermediate (grey), and hillslope (black) zones during the study period. Circles are mean values and error bars are standard deviations.

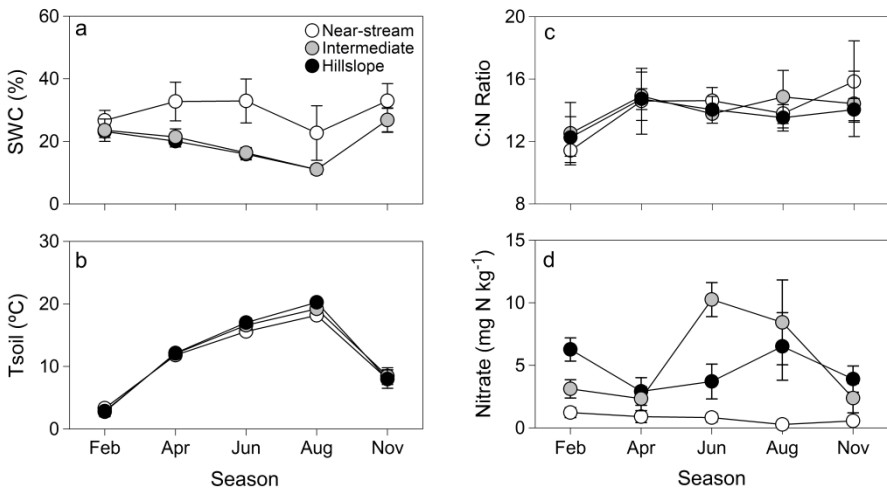




**Figure 3.** Temporal pattern of (a) soil net N mineralization (NNM), (b) net nitrification (NN) and (c) denitrification rates at the near-stream (white), intermediate (grey), and hillslope (black) zones during the study period. Bars are 610    mean values for each section and error bars are standard errors. For each season, different letters indicate significant differences among sections (mixed-model ANOVA, $p < 0.05$).

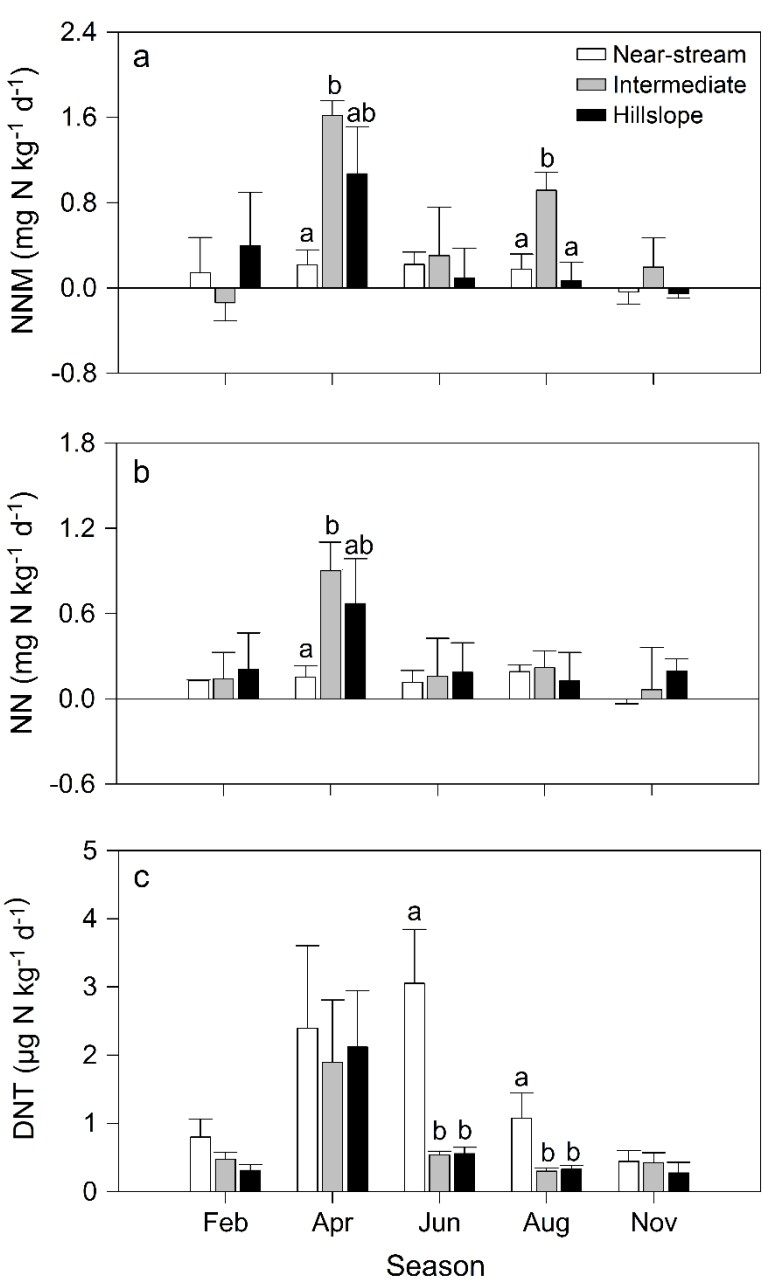



**Figure 4.** Temporal pattern of soil (a) $CO_2$ and (b) $N_2O$ emissions at the near-stream (white), intermediate (grey),
and hillslope (black) zones during the study period. Bars are mean values for each section and error bars are
standard errors. For each season, different letters indicate significant differences among sections (mixed-model
ANOVA, $p < 0.05$).

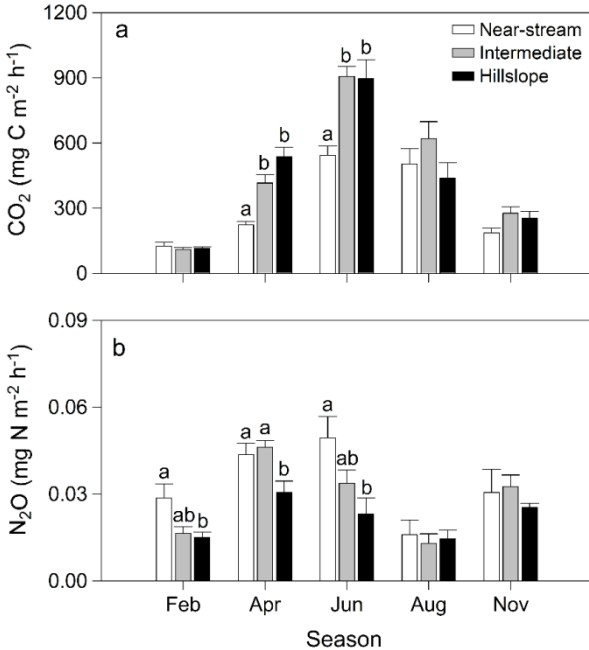





**Figure 5.** Loading plot of the (a) $CO_2$ and (b) $N_2O$ partial least squares models (PLS) for the 75 measurements. The graph depicts the correlation structures between the X variables (circles) and gas emissions (vectors). Variables situated along the same directional axis correlate with each other. Different color in X variables indicates their influence on gas emissions based on the "variable importance in the projection (VIP)" scores for each model. In each case, white has VIP scores $< 0.8$, grey has VIP scores $< 1.0$ and black has VIP scores $> 1.0$.

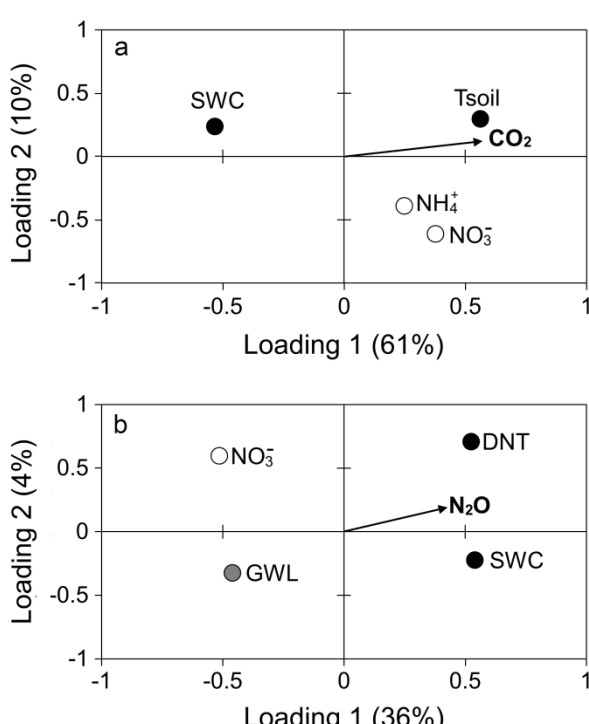
