# Peer review of "Temporal and spatial decoupling of CO2 and N2O soil emissions in a Mediterranean riparian forest"

_Biogeosciences, 2017_

## Referee Comment (RC1) · Anonymous Referee #1 · 6 Apr 2017

The manuscript title is not clear for the air emission or soil emission. The paper purpose needs to be sharpened. The details of CO2 and N2O measurement, emission calculation formulas, quality assurance, and statistical models should be given in method section. 1. In title: What is the CO2 and N2O soil emissions? It should be the CO2 and N2O emissions from soil. And in the context, "decoupling" is not discussed. 2. Row 69-80: one sentence is needed to clearly express the paper purpose. 3. Sections 2 and 3 may merge into one section as methodology section. 4. Can you give a schematic graph to show CO2 and N2O sampling locations in three zones and sampling schedules? 5. How did you measure CO2 in three zone at the same time with one sampling system? 6. Row 111: did you directly place the SRC-1 soil chamber on

top of the soil surface? 7. What standard operation procedures did you follow for CO2 and N2O measurement? How did you conduct the QA/QC? Or are you sure that your measurements were accurate? 8. What is your CO2 and N2O emission calculation formulae? 9. Rows 113 to 114: "CO2 emissions rates were calculated as the amount of CO2 accumulated in the head-space of the EGM-4 chamber after an incubation time of c.a. 120 s." Please make sure that is EGM-4 chamber or SRC-1 soil chamber? 10. In 3.3 section: can you give statistical model for each analysis 11. Can you give a graph to show CO2 and N2O concentrations or emission during June measurement with time to show real measurement? 12. How do you conclude the decoupling of CO2 and N2O emission from soils in time and location? 13. In conclusion, you may report your own conclusions. Why did you cite the references in your conclusion? What is your meaning about the large emission of GHG?

---

## Referee Comment (RC2) · Anonymous Referee #2 · 17 Apr 2017

In this paper entitled "Temporal and spatial decoupling of CO2 and N2O soil emissions in a Mediterranean riparian forest" Poblador et al report annual and seasonal greenhouse gas (GHG) emissions from a Spanish riparian zone. The authors found that N2O fluxes from denitrification were lower than in previous studies but that CO2 flux was quite high. As expected, these fluxes were negatively correlated in space and time. The authors conduct a sound observational study with interesting results, but I was left wondering what the key findings were and how they advanced our understanding of the role of riparian zones as a terrestrial-atmospheric-aquatic interface. I outline a few general concerns below before providing line edits. 1. The manuscript focuses on GHG emissions, but the temporal and spatial scale and the methods of the study

do not seem suited to answer this question. High CO2 fluxes in themselves do not indicate whether an ecosystem is a carbon source or sink, since net ecosystem carbon balance is the relevant parameter. Furthermore, there is large inter-annual variability in both CO2 and N2O fluxes in Mediterranean ecosystems, and the magnitude of difference observed here does not seem strong enough to infer landscape functions. I think the study has many other interesting implications about the link between microbial, physicochemical, and hydrological variation, but I feel it does not shine when framed as an assessment of GHG budgets. 2. The introduction reports many interesting observations but the lack of a focused research question, hypothesis, and broader conceptual framework make it hard to identify the salient points. Revising the intro to focus on clear question (rather than just stating multiple times that little is known about GHG flux from Mediterranean riparian zones) would strengthen the paper substantially. 3. The discussion currently feels like a continuation or repetition of the results sections. Clearly summarizing the key findings and their implications at the beginning of the discussion would orient the reader to better appreciate the value of this study. Subheadings could be effective at organizing the content and an overall shortening is probably in order since the discussion is quite long for the amount of new material it presents. 4. A conceptual figure laying out the expected or observed functioning of the riparian zone in regards to respiration and denitrification would be useful and could help focus the paper. 5. There are many unnecessary acronyms that make the text unwieldy. Avoiding uncommon acronyms (e.g. GWL, SWC, NNM, NN, DNT, PLS, DEA, TCD) would make the paper more accessible. 6. The paper is generally well written but it has quite a few non-standard phrasings and English formulations. Asking for a proofread from a native speaker would be worthwhile.

If the paper can be restructured around a compelling question, it could be a valuable contribution to our understanding of riparian zones in the larger landscape context, but the current lack of focus limits the paper in its current state.

Line edits: 21: powerless is a strange word choice 38: Not clear what 70% of total

emissions means. Also no need to put greenhouse in quotes 40: Not clear what this means? 44: contribute to increasing 59: may complicate upscaling (instead of may difficult to upscale) 72: alter instead of vary and measure instead of measured 88: compositions 111: with instead of by using 262: There is a large body of research on scaling riparian soil measurements. References: There are several inconsistently formatted references. 620: Figure 5 has a pretty low information content. I wonder if it could be included in the supplementary information.

---

## Referee Comment (RC3) · Anonymous Referee #3 · 18 Apr 2017

The paper is interesting, methods reliable and results almost as expected. The main concern is missing of data on potential N2 emission which is the main product of denitrification. Therefore, I cannot agree with the statement in paper that low N2O emission is a result of low intensity denitrification. In opposite, it could be that the denitrification process is complete and most of N2O produced will be transformed to N2. However, without evidences on (potential) N2 emission (either based on 15N or He-O2 analysis or even the acetylene method which gives underestimated but at least some values) and denitrification control genes (nirS+nirK and nosZi+II) it is hard to say about the intensity of denitrification. It can also be that a part of N2O is coming from nitrification. This kind of discussion is missing and may be it is too much to require analysis

of all those components. However, authors should avoid to declare that denitrification intensity is low because the N2O flux is low. Also, it is recommended to include some relevant references on denitrification intensity (N2:N2O ratio) in riparian zones and develop a short discussion based on this knowledge.

---

## Author Comment (AC1) · 2 Jun 2017

**Author responses to reviewer comments on the manuscript entitled**

**"Temporal and spatial decoupling of CO$_2$ and N$_2$O soil emissions in a Mediterranean riparian forest"**

**by Sílvia Poblador, Anna Lupon, Santiago Sabaté and Francesc Sabater**

Dear reviewers and Prof. Akihiko Ito,

Many thanks for your review and your positive and constructive comments on the manuscript "Temporal and spatial decoupling of CO$_2$ and N$_2$O soil emissions in a Mediterranean riparian forest". We have considered all the reviewers' comments and we are working thoroughly on a new version of the manuscript to tackle them all. As suggested by the reviewers, we are rewriting the introduction in order to clarify our objectives and improve the justification of our study design. In the new version, we will improve the methodology description, include the suggested new figures, and provide estimates of the N$_2$:N$_2$O ratios. Finally, we will shorten the discussion and we will tone down our conclusions. We feel that the reviewers' suggestions have helped to improve the quality and clarity of our manuscript and we believe that you will find our revised version suitable for publication in *Biogeosciences*.

Below we provide the specific answers to all the questions raised by the three reviewers. Please, do not hesitate to contact us if you considered that further clarifications are needed.

Looking forward hearing from you soon,

Sílvia Poblador,

Departament de Biologia Evolutiva, Ecologia i Ciències Ambientals, Universitat de Barcelona

Diagonal 643, 08028, Barcelona, Spain

cc: Anna Lupon, Santiago Sabaté, Francesc Sabater

**Reviewer #1**

General comments:

*The manuscript title is not clear for the air emission or soil emission. The paper purpose needs to be sharpened. The details of $CO_2$ and $N_2O$ measurement, emission calculation formulas, quality assurance, and statistical models should be given in method section.*

**Answer:** We are very grateful for the many insightful comments and suggestions. We agree that the title and the objectives of the older version of the manuscript were confusing. In the new version of the manuscript, we will change the title to clarify that we measure soil emissions. In addition, we will rewrite the last paragraph of the introduction by including the constructive comments from the reviewer and Reviewer 2, who also highlight the need to improve the purpose and hypotheses of the study. Finally, and following the reviewer suggestion, we will include detailed information of the methods and calculations used to estimate carbon dioxide ($CO_2$) emissions.

On the other hand, we have carefully checked all the calculations to ensure that there were no mistakes. By doing so, we realized that Figure 4a was wrong: the bars represent mg $CO_2$ m$^{-2}$ h$^{-1}$ (not mg C m$^{-2}$ h$^{-1}$). We apologize for the mistake and we are thankful to the reviewer to pointing it out. For more detail on the changes we have made to the manuscript, please see our responses below.

Detailed comments:

*1. In title: What is the $CO_2$ and $N_2O$ soil emissions? It should be the $CO_2$ and $N_2O$ emissions from soil. And in the context, "decoupling" is not discussed.* **Answer:** We agree that we did not use "*decoupling*" in a proper way; we meant that $CO_2$ and $N_2O$ had opposite spatial patterns. To avoid further confusions, and following the reviewer advice, we will change the title to: "Soil moisture drives spatiotemporal patterns of $CO_2$ and $N_2O$ soil emissions in a Mediterranean riparian forest".

*2. Row 69-80: one sentence is needed to clearly express the paper purpose.* **Answer:** The reviewer is right; we admit that our objectives and hypotheses were not clear enough. The main objectives of our study were (i) to evaluate the spatiotemporal patterns of $CO_2$ and $N_2O$ emissions and (ii) to analyze under which conditions soil moisture rules microbial processes and GHG emissions over other environmental constrains. These objectives will be specify in the new version of the manuscript.

*3. Sections 2 and 3 may merge into one section as methodology section.* **Answer:** OK, we will include the study site in the "Materials and methods" section.

*4. Can you give a schematic graph to show $CO_2$ and $N_2O$ sampling locations in three zones and sampling schedules?* **Answer:** We agree with the reviewer that a map of the studied riparian forest would improve the readability and justification of the study design. Accordingly, we will include a new figure showing the sampling locations along the studied riparian forest (Figure R.1).

[Figure]

*A. glutinosa*  *P. nigra*  *R. pseudoacacia*  *F. excelsior*

**Fig. R1:** Plot layout for the studied Mediterranean riparian forest showing the three riparian zones and the location of the chambers (n=5 for each riparian zone).

*5. How did you measure $CO_2$ in three zones at the same time with one sampling system?* **Answer:** We acknowledge that the explanation regarding $CO_2$ measurements was confusing. As the reviewer pointed, soil $CO_2$ effluxes were not measured simultaneously at all sites; instead, we used one unique sensor to perform all the measurements. Every day, at 12 pm, we started measuring soil $CO_2$ effluxes in the first site of the near-stream zone. When finished, we moved to the second site and so on until finished with all sites (15 in total). The incubation time needed for $CO_2$ efflux measurements was generally short (< 2 min), and thus, we were able to perform all the measurements in a relatively short time (~ 30 min). This field protocol ensured that changes in $CO_2$ emissions were because of the spatial variability rather than to diurnal changes in environmental conditions. We will clarify this in the new version of the manuscript.

*6. Row 111: did you directly place the SRC-1 soil chamber on top of the soil surface?* **Answer:** No, the SRC-1 chamber was gently placed on the top soil surface after (i) carefully removing the litter layer and (ii) checking that there was no air exchange with the atmosphere. Furthermore, prior to each measurement, we aerated the sensor (1 min) to minimize the contamination between samples. This procedure will be detailed in the new version of the manuscript to avoid confusions.

*7. What standard operation procedures did you follow for $CO_2$ and $N_2O$ measurement? How did you conduct the QA/QC? Or are you sure that your measurements were accurate?* **Answer:** For $CO_2$ measurements, we assumed an accuracy of 1% (PP Systems, USA). Moreover, the EGM-4 performed an "auto-zero" at regular intervals during the sampling to ensure the stability of $CO_2$ signals, minimize contamination, and reduce changes on sensitivity. For $N_2O$ measurements, we ensure the accuracy of the gas chromatograph measurements (Agilent Technologies, 7820A GC System) by measuring several certified standards (4.66 ppm $N_2O$; AirLiquide) prior, during, and post the analysis of our samples. Following your advice, we will provide a detail description of these procedures in the new methods section.

*8. What is your CO₂ and N₂O emission calculation formulae?* **Answer:** The rates of $CO_2$ and $N_2O$ emission were calculated from a best fit linear or exponential regression of gas concentration with time. GHG fluxes on an areal basis ($Fg$, in $\mu mol\ m^{-2}\ h^{-1}$) were then calculated following Healy et al. (1996):

$$F_g = \frac{dg}{dt} \times \frac{V\ P_0}{S\ RT_0} \qquad\qquad (Eq.1)$$

where dg/dt is the rate of change in gas concentration (in $\mu mol\ mol^{-1}\ h^{-1}$) in the chamber, $V$ is chamber volume (in $m^3$), $P_0$ is initial pressure (in Pa), S is the soil surface area (in $m^2$), R is the gas constant ($8.314\ Pa\ m^3\ K^{-1}\ mol^{-1}$), and $T_0$ is the initial chamber temperature (in °K). For budgeting, moles of $CO_2$ and $N_2O$ were converted to grams of C and N, respectively. Following the reviewer suggestion, we will include the formula in the new of the manuscript.

*9. Rows 113 to 114: "CO₂ emissions rates were calculated as the amount of CO₂ accumulated in the head-space of the EGM-4 chamber after an incubation time of c.a. 120 s." Please make sure that is EGM-4 chamber or SRC-1 soil chamber?* **Answer:** The reviewer had a point; this sentence was confusing. We will clarify in the text that is a SRC-1 soil chamber connected to an EGM-4.

*10. In 3.3 section: can you give statistical model for each analysis* **Answer:** We are not sure which statistical model the reviewer is referring to. However, in the new manuscript we will clarify that (i) we used the best linear or exponential fit model to estimate gas emission rates, (ii) we performed linear mixed-model analysis of variance (ANOVA) to test differences in soil properties, microbial N processes, and gas emissions across riparian zones and seasons and (iii) we used partial least squares regression (PLS) to explore how soil properties and soil N processes predict variation in $CO_2$ and $N_2O$ emissions. Moreover, Table 2 and Table 4 will show the main statistical results of the linear mixed-model ANOVA and the PLS-models, respectively. We will also include a new figure in the Supplementary Information showing the models used to estimated gas emissions rates (please, see comment 11 for more information).

*11. Can you give a graph to show CO₂ and N₂O concentrations or emission during June measurement with time to show real measurement?* **Answer:** Following the reviewer suggestion, Figure R.2 shows the increments of $CO_2$ and $N_2O$ concentrations during the incubation time for the field campaign of June 2013. The best fit linear regression used to estimate gas effluxes is shown in each case. Based on these regressions, rates of $CO_2$ and $N_2O$ emissions were 99.11 – 160.98 mg C m$^{-2}$ h$^{-1}$ and 0.0007 – 0.002 mg N m$^{-2}$ h$^{-1}$. Figure R2 will be included in the Supplementary Information of the manuscript.

[Figure]

**Fig. R2:** Concentrations of carbon dioxide (left column) and nitrous oxide (right column) during the incubation time for the sampling campaign of June 2013. Data is shown for the near-stream, intermediate and hillslope zones separately. For each plot, data is shown as mean ± SD (n = 5) for all sampling days of June. The best fit linear model used to calculate gas emissions is shown for each plot.

*12. How do you conclude the decoupling of $CO_2$ and $N_2O$ emission from soils in time and location?* **Answer:** As we acknowledged in your earlier comment, we did not used properly the term **"*decoupling*"**. By decoupling we meant that $CO_2$ and $N_2O$ fluxes were negatively correlated in space: $CO_2$ fluxes were higher at the hillslope zone, while higher $N_2O$ emissions were observed at the near-stream zone. Similarly, the temporal pattern of $CO_2$ and $N_2O$ emissions also differed: $CO_2$ emissions were maxima in June, while highest $N_2O$ emissions occurred in April. In any case, we will remove the term "*decoupling*" to avoid confusions.

*13. In conclusion, you may report your own conclusions. Why did you cite the references in your conclusion? What is your meaning about the large emission of GHG?* **Answer:** Following the reviewer suggestion, we have removed all citations from the conclusion section to avoid confusions and improve the readability of our findings and their meaningful. For instance, we do now explain that our results clearly illustrate the close linkage between riparian soil water availability and the microbial processes that produce GHG. Overall, these findings highlight that future variations in soil water availability due to climate change can potentially affect the riparian functionality in Mediterranean zones, as well as their contribution to regional and global C and N cycles.

**Reviewer #2**

*In this paper entitled "Temporal and spatial decoupling of $CO_2$ and $N_2O$ soil emissions in a Mediterranean riparian forest" Poblador et al report annual and seasonal greenhouse gas (GHG) emissions from a Spanish riparian zone. The authors found that $N_2O$ fluxes from denitrification were lower than in previous studies but that $CO_2$ flux was quite high. As expected, these fluxes were negatively correlated in space and time. The authors conduct a sound observational study with interesting results, but I was left wondering what the key findings were and how they advanced our understanding of the role of riparian zones as a terrestrial-atmospheric-aquatic interface. I outline a few general concerns below before providing line edits. If the paper can be restructured around a compelling question, it could be a valuable contribution to our understanding of riparian zones in the larger landscape context, but the current lack of focus limits the paper in its current state.*

**Answer:** Thank you for your positive and constructive comments, which have helped us to improve the manuscript significantly. Following your advice, we will carefully rewrite both the introduction and discussion sections in order to (i) develop a more meaningful objectives and hypotheses and (ii) highlight the novelties of our study. For more detail of the changes we will made to the manuscript, please see our responses below.

Major concerns:

*1. The manuscript focuses on GHG emissions, but the temporal and spatial scale and the methods of the study do not seem suited to answer this question. High $CO_2$ fluxes in themselves do not indicate whether an ecosystem is a carbon source or sink, since net ecosystem carbon balance is the relevant parameter. Furthermore, there is large inter-annual variability in both $CO_2$ and $N_2O$ fluxes in Mediterranean ecosystems, and the magnitude of difference observed here does not seem strong enough to infer landscape functions. I think the study has many other interesting implications about the link between microbial, physicochemical, and hydrological variation, but I feel it does not shine when framed as an assessment of GHG budgets.*

**Answer:** Thank you for your remarks. As we designed this study, we initially set out to address the relationship between soil water availability, microbial processes and GHG emissions in a Mediterranean riparian forest. Later on, the editor pinpoint that our results could be used to understand the role of riparian zones as active components of the regional and global carbon (C) and nitrogen (N) cycle. We agreed with him because our findings suggest that, on annual terms, Mediterranean riparian zones can transform and outgas large amounts of C and N. However, finding a balance between explaining the rationale of the study (which clearly involves the link between water availability, microbial transformations, and gas emissions) and focusing on more interesting implications of our results has proven challenging. We agree that the previous version of the manuscript may have put too much emphasis on the magnitude of greenhouse gas emissions. Therefore, we will make changes according to the reviewer suggestions (which we believe that will improve the logical structure of the introduction), while maintaining a mention of the C and N annual emissions in the discussion. Moreover, we also acknowledge that we cannot state if riparian zones are sink or source of C because we did not take into account C uptake rates. We will omit that concept in the new version of the manuscript.

*2. The introduction reports many interesting observations but the lack of a focused research question, hypothesis, and broader conceptual framework make it hard to identify the salient points. Revising the intro to focus on clear question (rather than just stating multiple times that little is known about GHG flux from Mediterranean riparian zones) would strengthen the paper substantially.*

**Answer:** Thanks for the suggestion; we will rewrite the introduction to improve the readability of our objectives and the justification of our study design. First, we will specify that our goals were: (i) to evaluate the spatiotemporal patterns of $CO_2$ and $N_2O$ emissions and (ii) to analyze under which conditions soil moisture rules microbial processes and GHG emissions over other environmental constrains. These objectives will be specify in the new version of the manuscript.

Further, we will improve our hypotheses by focusing on the role of soil water availability on microbial processes. We hypothesized that the magnitude and the relative contribution of $N_2O$ and $CO_2$ to total GHG emission would strongly depend on soil moisture. In the near-stream zone, we expected that saturated soils would enhance denitrification and methanogenesis, but inhibited both respiration and nitrification. Thus, we predicted higher $N_2O$ than $CO_2$ emissions in this zone. In the intermediate zone, we expected that wet (but not saturated) soils would enhance aerobic processes, and thus, we predicted high $CO_2$ emissions compared to $N_2O$. Finally, we expected that dry soils would depleted (or even inhibited) microbial activity near the hillslope edge, and therefore, we predicted low GHG emissions in this zone. Because Mediterranean regions are subjected to strong intra-annual variations in moisture and temperature, we predicted that this

general behavior may be maximized during summer, when only near-stream soils would keep wet. Conversely, differences would be minimized during wet periods, where all zones would have high rates of GHG emissions.

Finally, we will rewrite the introduction to focus more on the relationship between soil water availability, microbial processes and GHG emission rather than in the source/sink behavior of riparian forests in the global C and N cycle. Moreover, we will highlight that Mediterranean systems are a unique natural laboratory to understand the close link between water availability and riparian functionality because (i) they are characterized by a marked spatial gradient in soil moisture and (ii) they are subjected to seasonal alterations of precipitation and temperature regimes that might affect soil water availability in riparian soils. With these improvements, we strongly believe that the introduction will be better framed within the context of riparian functionality and will explicitly address the rational and underlying hypotheses that were implicit in our study design.

*3. The discussion currently feels like a continuation or repetition of the results sections. Clearly summarizing the key findings and their implications at the beginning of the discussion would orient the reader to better appreciate the value of this study. Subheadings could be effective at organizing the content and an overall shortening is probably in order since the discussion is quite long for the amount of new material it presents.*

**Answer:** We agreed that the discussion is quite long and repetitive. Following the reviewer advice, we will open the discussion with a short summary highlighting the main objectives and results of the study. Moreover, we will include new subsections and reduce the redundancy between them. The new discussion will have four subsections that will discuss the main results regarding (i) the microbial processes that regulated GHG emissions (section 4.1), (ii) the effect of soil water availability on $CO_2$ and $N_2O$ emissions (sections 4.2 and 4.3, respectively), and (iii) the potential role of riparian soils as a hot spot of GHG emissions (section 4.4). We believe that these changes, together with those proposed for the other reviewers, will reduce significantly the length of the discussion as well as will better highlight the implications of our findings.

*4. A conceptual figure laying out the expected or observed functioning of the riparian zone in regards to respiration and denitrification would be useful and could help focus the paper.*

**Answer:** As suggested, we performed a conceptual figure showing the expected link between water availability and microbial processes along our riparian forest (Figure R.3). Denitrification and methanogenesis are anaerobic processes that only occur under anoxic conditions (water content > 60%), while respiration, mineralization and nitrification are aerobic process that mostly occur under moist soils (water content = 30-60%) (Pinay et al., 2007). Based on these premises, we hypothesized that soil moisture would control the microbial processes and GHG emissions in our Mediterranean riparian soils. We expected high denitrification rates in the near-stream zone, where high groundwater levels sustain saturated soil. Conversely, we expected high respiration and mineralization rates in the intermediate zone, where soils are wet but not saturated. Finally, we expected low (or nil) microbial activity in the hillslope zone, where deep groundwater tables are not sufficient to keep the top soils moist.

Although this figure may help to understand some of our expectations and predictions, we are confident that our new hypotheses are much better explained now, and therefore, we decided not to include it in the main manuscript.

[Figure]

**Fig. R3.** Conceptual model of the influence of hydrology on soil microbial processes across a Mediterranean riparian zone. Soil moisture decreases from the near-stream to the hillslope zones due to changes in groundwater table, increasing unsaturated soil column and oxic conditions. Anaerobic processes such as denitrification and methanogenesis occur under anoxic conditions, and thus, high rates of these processes are expected in the near-stream zone. Conversely, aerobic processes such as respiration and mineralization are optimized under a moderate range of moisture, and therefore, they would increase in the intermediate zone. Finally, we expected low (or nil) microbial activity in the hillslope zone because dry soils are unfavorable for both aerobic and anaerobic processes. Accordingly, we predicted high rates of soil $CO_2$ effluxes in the intermediate zone, while high rates of $N_2O$ emissions are predicted in the near-stream zone.

*5. There are many unnecessary acronyms that make the text unwieldy. Avoiding uncommon acronyms (e.g. GWL, SWC, NNM, NN, DNT, PLS, DEA, TCD) would make the paper more accessible.*

**Answer:** OK; we will avoid the use of acronyms throughout the manuscript. However, we decided to keep them in the tables to improve their readability.

*6. The paper is generally well written but it has quite a few non-standard phrasings and English formulations. Asking for a proofread from a native speaker would be worthwhile.*

**Answer:** OK, a native English speaker will carefully read the new version of the manuscript before submission.

Detailed comments:

*Line 21: powerless is a strange word choice* **Answer:** OK, "low amounts of $N_2O$ emissions".

*Line 38: Not clear what 70% of total emissions means. Also no need to put greenhouse in quotes* **Answer:** We meant that riparian zones can emit up to 70% of global GHG emissions, which take into account both natural processes and human activities. We will clarify it in the text.

*Line 40: Not clear what this means?* **Answer:** We meant that soil water saturation can support large methane ($CH_4$) fluxes that account for the 15 – 40 % of the global $CH_4$ emissions. This sentence will be clarify in the text.

*Line 44: contribute to increasing* **Answer:** OK.

*Line 59: may complicate upscaling (instead of may difficult to upscale)* **Answer:** OK.

*Line 72: alter instead of vary and measure instead of measured* **Answer:** OK.

*Line 88: compositions* **Answer:** OK.

*Line 111: with instead of by using.* **Answer:** OK.

*Line 262: There is a large body of research on scaling riparian soil measurements.* **Answer:** That's right; several studies have attempted to upscale soil processes and how the mechanisms underlying such GHG emissions can ultimately modify catchment GHG fluxes (e.g. Hagedorn, 2010; Pinay et al., 2015; Vidon and Hill, 2006). However, there are still fundamental uncertainties regarding the magnitude, spatiotemporal variation, and sources of GHG emissions from riparian zones (see Pinay et al., 2015). We will clarify it in the new version of the manuscript.

*References: There are several inconsistently formatted references.* **Answer:** OK, references will be carefully checked before submitting the new manuscript.

*Line 620: Figure 5 has a pretty low information content. I wonder if it could be included in the supplementary information.* **Answer:** OK, Figure 5 will be moved to supplementary information.

**Reviewer #3**

*The paper is interesting, methods reliable and results almost as expected. The main concern is missing of data on potential $N_2$ emission which is the main product of denitrification. Therefore, I cannot agree with the statement in paper that low $N_2O$ emission is a result of low intensity denitrification. In opposite, it could be that the denitrification process is complete and most of $N_2O$ produced will be transformed to $N_2$. However, without evidences on (potential) $N_2$ emission (either based on 15N or He-$O_2$ analysis or even the acetylene method which gives underestimated but at least some values) and denitrification control genes (nirS+nirK and nosZi+II) it is hard to say about the intensity of denitrification. It can also be that a part of $N_2O$ is coming from nitrification. This kind of discussion is missing and may be it is too much to require analysis of all those components. However, authors should avoid to declare that denitrification intensity is low because the $N_2O$ flux is low. Also, it is recommended to include some relevant references on denitrification intensity ($N_2$:$N_2O$ ratio) in riparian zones and develop a short discussion based on this knowledge.*

**Answer:** Certainty, measures of denitrification intensity (i.e. $N_2$:$N_2O$ ratio) can be an added value to our discussion. Unfortunately, we did not measure $N_2$ nor denitrification control genes to identify the source of $N_2O$ emissions (Butterbach-bahl et al., 2013; Mander et al., 2014; Saarenheimo et al., 2015). However, we indeed measured denitrification rates by the acetylene method. Therefore, as the reviewer pinpointed, we can calculate some rough estimates of $N_2$:$N_2O$ ratio. As shown in Table R.1, $N_2$:$N_2O$ ratios ranged from 4 (hillslope zone) to 22 (near-stream zone), while some studies in temperate riparian forests reported N2:N2O ratios of $184 - 844$ (Mander et al., 2014). These results support the idea that low $N_2O$ emissions in our riparian site can be attributed to both low denitrification rates (0.72 - 2.69 mg N $kg^{-1}$ $d^{-1}$) and high denitrification efficiency. The highest denitrification efficiency was observed in the near-stream zone, suggesting that saturated soils favored the complete denitrification process and can potentially emit less $N_2O$ compared to less saturated soils. Following the reviewer suggestion, we will include this rationale in the new discussion; thanks for pointing it out.

**Table R.1:** Mean values (± standard deviation) of $N_2$ emissions, $N_2O$ emissions and molar $N_2$:$N_2O$ ratios for each riparian zone during the study period.

| | $N_2$ Emission (mg m$^{-2}$ d$^{-1}$) | $N_2O$ Emission (mg m$^{-2}$ d$^{-1}$) | $N_2$:$N_2O$ ratio |
|---|---|---|---|
| **Near-stream** | 4.93 ± 10.17 | 0.80 ± 0.79 | 21.50 ± 40.32 |
| **Intermediate** | 0.41 ± 1.41 | 0.79 ± 0.81 | 5.90 ± 16.02 |
| **Hillslope** | 0.42 ± 1.05 | 0.52 ± 0.52 | 4.23 ± 8.31 |

**References**

Butterbach-bahl, K., Baggs, E. M., Dannenmann, M., Kiese, R. and Zechmeister-boltenstern, S.: Nitrous oxide emissions from soils : how well do we understand the processes and their controls ?, Philos. Trans. R. Soc., 2013.

Hagedorn, F.: Hot spots and hot moments for greenhouse gas emissions from soils, Swiss Fed. Inst. For. Snow Landsc. Res., (1), 2010.

Healy, R. W., Striegl, R. G., Russell, T. F., Hutchinson, G. L. and Livingston, G. P.: Numerical Evaluation of Static-Chamber Measurements of Soil - Atmosphere Gas Exchange : Identification of Physical Processes, Soil Sci. Soc. Am. J., 60, doi:10.2136/sssaj1996.03615995006000030009x, 1996.

Mander, Ü., Well, R., Weymann, D., Soosaar, K., Maddison, M., Kanal, A., Lõhmus, K., Truu, J., Augustin, J. and Tournebize, J.: Isotopologue Ratios of $N_2O$ and $N_2$ Measurements Underpin the Importance of Denitrifi cation in Differently N‑Loaded Riparian Alder Forests, Environ. Sci. Technol., 48, 11910–11918, doi:dx.doi.org/10.1021/es501727h, 2014.

Pinay, G., Gumiero, B., Tabacchi, E., Gimenez, O., Tabacchi-Planty, a. M., Hefting, M. M., Burt, T. P., Black, V. a., Nilsson, C., Iordache, V., Bureau, F., Vought, L., Petts, G. E. and Décamps, H.: Patterns of denitrification rates in European alluvial soils under various hydrological regimes, Freshw. Biol., 52(2), 252–266, doi:10.1111/j.1365-2427.2006.01680.x, 2007.

Pinay, G., Peiffer, S., De Dreuzy, J.-R., Krause, S., Hannah, D. M., Fleckenstein, J. H., Sebilo, M., Bishop, K. and Hubert-moy, L.: Upscaling Nitrogen Removal Capacity from Local Hotspots to Low Stream Orders ' Drainage Basins, Ecosystems, 18, 1101–1120, doi:10.1007/s10021-015-9878-5, 2015.

Saarenheimo, J., Rissanen, A. J., Arvola, L., Nykänen, H., Lehmann, M. F. and Tiirola, M.: Genetic and Environmental Controls on Nitrous Oxide Accumulation in Lakes, PLoS One, 10(3), doi:10.1371/journal.pone.0121201, 2015.

Vidon, P. G. and Hill, A. R.: A landscape-based approach to estimate riparian hydrological and nitrate removal functions, J. Am. Water Resour. Assoc., 3, 1099–1112, 2006.